# SPI-GAN: DENOISING DIFFUSION GANS WITH STRAIGHT-PATH INTERPOLATIONS

## ABSTRACT

Score-based generative models (SGMs) show the state-of-the-art sampling quality and diversity. However, their training/sampling complexity is notoriously high due to the highly complicated forward/reverse processes, so learning a simpler process is gathering much attention currently. We present an enhanced GAN-based denoising method, called SPI-GAN, using our proposed *straight-path interpolation* definition. To this end, we propose a GAN architecture i) denoising through the straight-path and ii) characterized by a continuous mapping neural network for imitating the denoising path. This approach drastically reduces the sampling time while achieving as high sampling quality and diversity as SGMs. As a result, SPI-GAN is one of the best-balanced models among the sampling quality, diversity, and time for CIFAR-10, CelebA-HQ-256, and LSUN-Church-256.

## 1 INTRODUCTION

Generative models are one of the most popular research topics for deep learning. Many different models have been proposed, ranging from variational autoencoders (Kingma & Welling, 2013) and generative adversarial networks (Goodfellow et al., 2014) to recent denoising diffusion models (Ho et al., 2020; Song & Ermon, 2019; Song et al., 2021c). The representative denoising diffusion models score matching with Langevin dynamics (Song & Ermon, 2019) and denoising diffusion probabilistic modeling (Ho et al., 2020) progressively corrupt original data and revert the corruption process to build a generative model. Recently, Song et al. (2021c) proposed a stochastic differential equation (SDE)-based mechanism that embraces all those models and coined the term, *score-based generative models* (SGMs).

Each generative model has different characteristics in terms of the generative task trilemma: i) sampling quality, ii) sampling diversity, and iii) sampling time. Generative adversarial networks (GANs) generate samples with high quality but low diversity. Conversely, variational autoencoders (VAEs) generate a variety of samples, but its sampling quality is lacking. SGMs outperform GANs and VAEs in terms of sampling quality/diversity. However, sampling with SGMs takes a lot of time.

Resolving the trilemma of generative models is an important recent research topic. In particular, reducing the complexity of SGMs is gathering much attention. There exist two different directions for this: i) learning a simpler process than the complicated forward/reverse process of SGMs (Nichol & Dhariwal, 2021; Das et al., 2023), and ii) letting GANs imitate [1] SGMs (Xiao et al., 2021; Wang et al., 2022). Our method can be considered a hybrid of them. Inspired by the previous studies, we propose a GAN-base hybrid method that approximates the straight-path interpolation. Our method imitates a denoising process following the straight-path interpolation guided by $u\mathbf{x}_0 + (1 - u)\mathbf{x}_T$, where $0 \leq u \leq 1$ (cf. Figure 1 (d)). Therefore, we call our method *straight-path interpolation GAN* (SPI-GAN).

One may consider that SPI-GAN is similar to Denoising Diffusion GAN (DD-GAN) and Diffusion-GAN (cf. Figure 1 (b-d)). DD-GAN (Xiao et al., 2021) effectively reduces the number of denoising steps by letting its GAN approximate the shortcuts. In Diffusion-GAN (Wang et al., 2022), image

---

[1] Let $\mathbf{x}_0$ be a clean original sample and $\mathbf{x}_T$ be a noisy sample that follows a Gaussian prior under the context of SGMs. These imitation methods learn a denoising process following the reverse SDE path by training their conditional generators to read $\mathbf{x}_t$ and output $\mathbf{x}_{t-j}$. Typically, a large $j > 0$ is preferred to reduce the denoising step (see Section 2 for a detailed explanation).

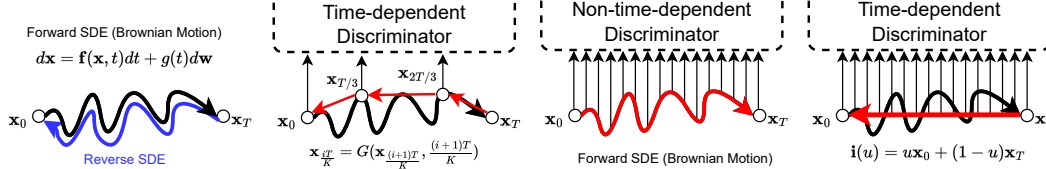

(a) The SDE-based work-flow of SGMs. The reverse SDE is a generation process.

(b) Approximating the reverse SDE ($T$ steps in total) with $K$ shortcuts by DD-GAN, e.g., $K = 3$ in this example.

(c) Augmentation following the forward SDE of Diffusion-GAN.

(d) Our proposed denoising method, SPI-GAN, with straight-path interpolations.

Figure 1: The comparison among four models: i) the original formulation of SGMs in (a), ii) DD-GAN's learning the shortcuts of the reverse SDE (based on the conditional generator $\mathbf{x}_{\frac{iT}{K}} = G(\mathbf{x}_{\frac{(i+1)T}{K}}, \frac{(i+1)T}{K})$) in (b), iii) Diffusion-GAN's augmentation method in (c) and iv) SPI-GAN's learning the straight path in (d). The red paths in (b), (c), and (d) are used as the discriminators' input. Note that we do not strictly follow the reverse SDE path but the straightly interpolated path. Therefore, one can consider that SPI-GAN denoises $\mathbf{x}_T$ to $\mathbf{x}_0$ following the straight path. The straight path is much easier to learn than the highly non-linear backward SDE path (see Section 2 and Appendix B for a more detailed discussion).

augmentation is performed by injecting noisy images following the forward path of SGMs, and the standard adversarial training is conducted for its GAN. However, our SPI-GAN is technically different and more sophisticated in the following points:

- Our straight-path interpolation is as simple as Eq. 6, is much easier to learn. It is also possible to derive its ordinary differential equation (ODE)-based formulation, which we call the straight-path interpolation process in Eq. 10 in Appendix B.

- In DD-GAN, the generator learns $K$ shortcuts through the reverse path of SGMs, whereas SPI-GAN learns the straight-interpolation path.

- Diffusion-GAN's discriminator is trained with the augmented noisy images without being conditioned on time — in other words, it is non-time-dependent. SPI-GAN's time-dependent discriminator learns the straight-path information, being conditioned on time.

- In order to learn a simple process, i.e., our straight-path interpolation, SPI-GAN uses a special neural network architecture, characterized by a mapping network.

- After all these efforts, SPI-GAN is a GAN-based method that imitates the straight-path interpolation. However, our mapping network is designed for this purpose, allowing SPI-GAN to generate fake images directly without recursion.

DD-GAN, Diffusion-GAN, and SPI-GAN attempt to address the trilemma of the generative model using GANs. Among those models, our proposed SPI-GAN, which learns a straight-path interpolation, shows the best balance in terms of the overall sampling quality, diversity, and time in three benchmark datasets: CIFAR-10, CelebA-HQ-256, and LSUN-Church-256.

## 2 RELATED WORK AND PRELIMINARIES

**Neural ordinary differential equations (NODEs).** Neural ordinary differential equations (Chen et al., 2018) use the following equation to define the continuous evolving process of the hidden vector $\mathbf{h}(u)$:

$$\mathbf{h}(u) = \mathbf{h}(0) + \int_0^u f(\mathbf{h}(t), t; \boldsymbol{\theta_f}) dt, \tag{1}$$

where the neural network $f(\mathbf{h}(t), t; \boldsymbol{\theta_f})$ learns $\frac{d\mathbf{h}(t)}{dt}$. To solve the integral problem, we typically rely on various ODE solvers. The explicit Euler method is one of the simplest ODE solvers. The 4th order

Runge–Kutta (RK4) method is a more sophisticated ODE solver and the Dormand–Prince (Dormand & Prince, 1980) method is an advanced adaptive step-size solver. The NODE-based continuous-time models (Chen et al., 2018; Kidger et al., 2020) show good performance in processing sequential data.

**Score-based generative models.** In diffusion models, the diffusion process is adding noises to a real image $\mathbf{x}_0 \sim q(\mathbf{x}_0)$ in $T$ steps as follows:

$$q(\mathbf{x}_{1:T}|\mathbf{x}_0) = \prod_{t \geq 1} q(\mathbf{x}_t|\mathbf{x}_{t-1}),$$

$$q(\mathbf{x}_t|\mathbf{x}_{t-1}) = \mathcal{N}(\mathbf{x}_t; \sqrt{1-\beta_t}\mathbf{x}_{t-1}, \beta_t \mathbf{I}), \tag{2}$$

where $\beta_t$ is a pre-defined variance schedule and $q(\mathbf{x}_0)$ is a data-generating distribution. The denoising (reverse) process of diffusion models is as follows:

$$p_{\boldsymbol{\theta}}(\mathbf{x}_{0:T}) = p(\mathbf{x}_T) \prod_{t \geq 1} p_{\boldsymbol{\theta}}(\mathbf{x}_{t-1}|\mathbf{x}_t),$$

$$p_{\boldsymbol{\theta}}(\mathbf{x}_{t-1}|\mathbf{x}_t) = \mathcal{N}(\mathbf{x}_{t-1}; \mu_{\boldsymbol{\theta}}(\mathbf{x}_t, t), \sigma_t^2 \mathbf{I}), \tag{3}$$

which $\boldsymbol{\theta}$ is denoising model's parameter, and $\mu_{\boldsymbol{\theta}}(\mathbf{x}_t, t)$ and $\sigma_t^2$ are the mean and variance for the denoising model. Afterward, score-based generative models (SGMs) generalize the diffusion process to continuous using SDE. SGMs use the following Itô SDE to define diffusive processes:

$$d\mathbf{x} = f(\mathbf{x}, t)dt + g(t)d\mathbf{w}, \tag{4}$$

where $\mathbf{w}$ is the standard Wiener process (a.k.a, Brownian motion), $f(\mathbf{x}, t)$ and $g(t)$ are defined in Appendix A. Following the Eq. 4, we can derive an $\mathbf{x}_t$ at time $t \in [0, T]$. As the value of $t$ increases, $\mathbf{x}_t$ approaches to $\mathcal{N}(\mathbf{0}, \sigma^2 \mathbf{I})$. The denoising process (reverse SDE) of SGMs is as follows:

$$d\mathbf{x} = [f(\mathbf{x}, t) - g^2(t)\nabla_{\mathbf{x}} \log p_t(\mathbf{x})]dt + g(t)d\mathbf{w}, \tag{5}$$

where $\nabla_{\mathbf{x}} \log p_t(\mathbf{x})$ is the gradient of the log probability density function. In the denoising process (i.e., the reverse SDE), a noisy sample at $t = T$ is mapped to a clean sample at $t = 0$. To overcome the slow sampling speed with a large $T$ (e.g., CIFAR-10: $T = 1000$) in SGMs, several approaches have been proposed, including i) learning an adaptive noise schedule (San-Roman et al., 2021), ii) introducing non-Markovian diffusion processes (Song & Ermon, 2020; Kong & Ping, 2021), iii) using faster SDE solvers for continuous-time models (Jolicoeur-Martineau et al., 2021a), iv) knowledge distillation (Luhman & Luhman, 2021), v) learning a simple path (Nichol & Dhariwal, 2021; Das et al., 2023) and vi) letting GANs imitate SGMs (Xiao et al., 2021; Zheng et al., 2022; Wang et al., 2022).

**Learning a simple path.** Deep generative models find mapping paths between data distributions and Gaussian prior distributions. There exist various definitions of *the best mapping path*. For instance, the Wasserstein distance calculates the distance between two distributions based on the optimal transport theory, i.e., mapping a particle from a prior distribution to a particle from a target distribution following the optimal transport path leading to the least action (Moosmüller & Cloninger, 2020). On the contrary, SGMs resort to much more complicated paths for mapping particles from the two distributions. Recently, learning a simpler path that shows faster training and sampling times, instead of the reverse SDE path in SGMs, is gathering much attention (Nichol & Dhariwal, 2021; Das et al., 2023). Das et al. (2023) suggested its own definition of the shortest mapping path in terms of the Fisher metric. However, these methods fail to show comparable sampling quality to our method since they are still in the infancy period of study.

For our proposed model, we resort to the shortest path in terms of the optimal transport path. To have our own shortest path definition, we use the straight-path interpolation from a Gaussian prior distribution to a target real image distribution. Wasserstein GANs (Arjovsky et al., 2017) implicitly learn optimal paths to minimize the Wasserstein distance whereas our model explicitly extracts straight-paths from SGMs, i.e., the two ends of a path are still defined by the SDE path of SGMs, and teaches them to the generator.

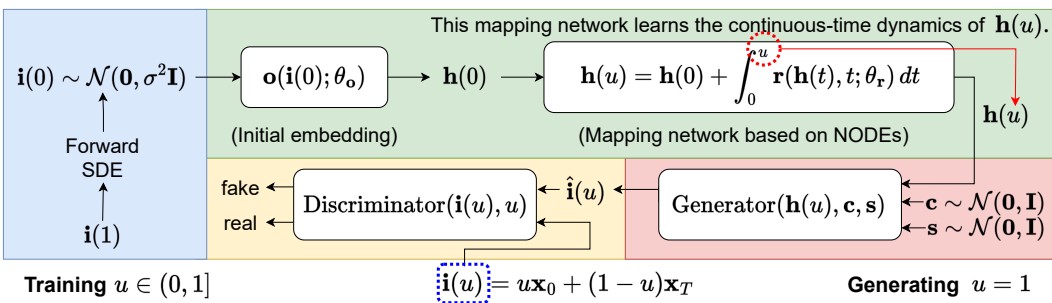

Figure 2: The architecture of our proposed SPI-GAN. $\mathbf{h}(u)$ is a latent vector which generates an interpolated image $\mathbf{i}(u)$ at time $u$. Therefore, $\mathbf{i}(1)$ is an original image and $\mathbf{i}(0)$ is a noisy image. We perform this adversarial training every time $u$ but generate images with $u = 1$. The constant noise $\mathbf{c}$ and the layer-wise varying noise $\mathbf{s}$ enable the stochasticity of the generator. Each color represents a computation step.

**Letting GANs imitate SGMs.** Among those enhancements of SGMs, letting GANs imitate SGMs shows the best-balanced performance in terms of the generative task trilemma. Figure 1 (b) shows the key idea of DD-GAN, which proposed to approximate the reverse SDE process with $K$ shortcuts. They internally utilize a GAN-based framework conditioned on time (step) $t$ to learn the shortcuts. A generator of DD-GAN receives noise images as input and denoises it. In other words, the generator generates clean images through denoising $K$ times from the noise images. To this end, DD-GAN uses a conditional generator, and $\mathbf{x}_t$ is used as a condition to generate $\mathbf{x}_{t-1}$. It repeats this $K$ times to finally creates $\mathbf{x}_0$. For its adversarial learning, the generator can match $p_{\boldsymbol{\theta}}(\mathbf{x}_{t-1}|\mathbf{x}_t)$ and $q(\mathbf{x}_{t-1}|\mathbf{x}_t)$. Figure 1 (c) shows the overall method of Diffusion-GAN. In contrast to DD-GAN, Diffusion-GAN directly generates clean images and its discriminator is trained after augmenting real images with noisy images that follow the forward SDE from the generated images. The noisy images are used as input to the discriminator to prevent mode-collapse in GANs.

## 3 PROPOSED METHOD

Our proposed method, SPI-GAN, learns how to denoise $\mathbf{x}_T$ to $\mathbf{x}_0$ following the straight-path during its training phase. After being trained, a latent vector $\mathbf{z} \sim \mathcal{N}(\mathbf{0}, \sigma^2\mathbf{I})$ is denoised to a fake image *directly without any recursion* (see Sections 3.8 and 4.4).

### 3.1 OVERALL WORKFLOW

We first describe the overall workflow of our proposed method. Before describing it, the notations in this paper are defined as follows: i) $\mathbf{i}(u) \in \mathbb{R}^{C \times H \times W}$ is an image with a channel $C$, a height $H$, and a width $W$ at interpolation point $u$. ii) $\hat{\mathbf{i}}(u) \in \mathbb{R}^{C \times H \times W}$ is a generated fake image at interpolation point $u$. iii) $\mathbf{h}(u) \in \mathbb{R}^{\dim(\mathbf{h})}$ is a latent vector at interpolation point $u$. iv) $\mathbf{r}(\mathbf{h}(t), t; \boldsymbol{\theta}_\mathbf{r}) \in \mathbb{R}^{\dim(\mathbf{h})}$ is a neural network approximating the time derivative of $\mathbf{h}(t)$, denoted $\frac{d\mathbf{h}(t)}{dt}$. Our proposed SPI-GAN consists of four parts, each of which has a different color in Figure 2, as follows:

1. **1st part (blue):** The first part, highlighted in blue in Figure 2, means that we calculate a noisy image $\mathbf{i}(0) = \mathbf{x}_T$ from $\mathbf{i}(1) = \mathbf{x}_0$ with the forward pass of the SGM. We note that this can be done in $\mathcal{O}(1)$.

2. **2nd part (green):** The second part maps a noisy vector $\mathbf{i}(0)$ into another latent vector $\mathbf{h}(u)$ after solving the integral problem. We note that $u$ is an interpolation point, which can be in $(0, 1]$. The final integral time of the NODE layer denoted $u$ with a red dotted circle, determines which latent vector it will generate.

3. **3rd part (red):** The third part is a generative step to generate a fake image $\hat{\mathbf{i}}(u)$ from $\mathbf{h}(u), \mathbf{c}, \mathbf{s}$ — the constant noise $\mathbf{c}$ and the layer-wise varying noise $\mathbf{s}$ are adopted from StyleGAN2 (Karras et al., 2020b). Our generator does not require $u$ as input, which means that $\mathbf{h}(u)$ internally has the temporal information. In other words, the latent space, where we sample $\mathbf{h}(u)$, is a common space across $u \in (0, 1]$.

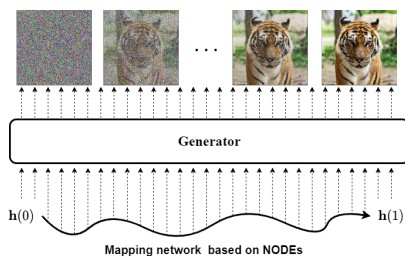

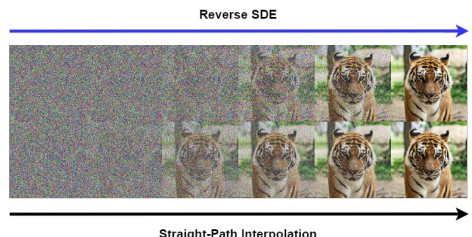

Figure 3: Process of generating sample from continuous latent vector.

Figure 4: Difference between reverse SDE and interpolation.

4. **4th part (yellow):** The fourth part is to distinguish between real and fake images (conditioned on $u$). Our discriminator discriminates not only clean images but also interpolated images. In other words, we maintain only one discriminator regardless of $u$.

5. After training with $u \in (0, 1]$, our generator is able to generate fake images $\mathbf{i}(1)$ directly.

## 3.2 DIFFUSION THROUGH THE FORWARD SDE

In the first part highlighted in blue, $\sigma^2$ can be different for $\mathbf{i}(0) \sim \mathcal{N}(\mathbf{0}, \sigma^2 \mathbf{I})$ depending on the type of the forward SDE. The type of SDE we consider is in Appendix A. Unlike the reverse SDE, which requires step-by-step computation, the forward SDE can be calculated with one-time computation for a target time $t$ (Song et al., 2021c). Therefore, it takes $\mathcal{O}(1)$ in the first part. We note that $\mathbf{i}(0) = \mathbf{x}_T$ and $\mathbf{i}(1) = \mathbf{x}_0$ are the two ends of the straight-path interpolation which will be described shortly.

## 3.3 STRAIGHT-PATH INTERPOLATION

SPI-GAN has advantages over SGMs since it learns a much simpler process than the SDE-based path of SGMs. Our straight-path interpolation (SPI) is defined as follows:

$$\mathbf{i}(u) = u\mathbf{x}_0 + (1 - u)\mathbf{x}_T, \tag{6}$$

where $u \in (0, 1]$ is an intermediate point and therefore, $\mathbf{i}(1) = \mathbf{x}_0$ and $\mathbf{i}(0) = \mathbf{x}_T$. We use a straight-path between $\mathbf{i}(1)$ and $\mathbf{i}(0)$ to learn the shortest path with the minimum Wasserstein distance. Therefore, Our straight-path interpolation process is simpler and thus more suitable for neural networks to learn. During our training process, we randomly sample $u$ every iteration rather than using fixed intermediate points. This random sampling method has advantages over utilizing fixed points for $u$ (cf. Section 4.3).

## 3.4 MAPPING NETWORK

The mapping network, which generates a latent vector $\mathbf{h}(u)$, is the most important component in our model. The mapping network consists of an initial embedding network, denoted $\mathbf{o}$, that generates the initial hidden representation $\mathbf{h}(0)$ from $\mathbf{i}(0)$, and a NODE-based mapping network. The role of network $\mathbf{o}$ is to reduce the size of the input to the mapping network for decreasing sampling time. In addition, the NODE-based mapping network generates the latent vector for a target interpolation point $u$, whose initial value problem (IVP) is defined as follows:

$$\mathbf{h}(u) = \mathbf{h}(0) + \int_0^u \mathbf{r}(\mathbf{h}(t), t; \boldsymbol{\theta}_\mathbf{r}) dt, \tag{7}$$

where $\frac{d\mathbf{h}(t)}{dt} = \mathbf{r}(\mathbf{h}(t), t; \boldsymbol{\theta}_\mathbf{r})$, and $\mathbf{r}$ has multiple fully-connected layers in our implementation. In general, $\mathbf{h}(0)$ is a lower-dimensional representation of the input. One more important point is that we maintain a single latent space for all $u$ and therefore, $\mathbf{h}(u)$ has the information of the image to generate at a target interpolation time $u$. For instance, Figure 3 shows that a noisy image is generated from $\mathbf{h}(0)$ but a clean image from $\mathbf{h}(1)$. As a matter of fact, NODEs are homeomorphic mappings over time and we exploit this characteristics to achieve our goal. We refer to Appendix C for the detailed explanation.

---

**Algorithm 1** How to sampling SPI-GAN

---

**Input**: Noisy vectors $\mathbf{z}^l \sim \mathcal{N}(\mathbf{0}, \sigma^2\mathbf{I})$

1: Sample a set of noisy vectors $\{\mathbf{z}^l\}_{l=1}^N$
2: Calculate $\{\mathbf{h}^l(1)\}_{l=1}^N$ via the mapping network which processes $\{\mathbf{z}^l\}_{l=1}^N$
   by solving Eq. 7 after setting $u = 1$
3: Generate a fake image $\{\hat{\mathbf{i}}^l(1)\}_{l=1}^N$ with the generator
4: **return** fake image $\{\hat{\mathbf{i}}^l(1)\}_{l=1}^N$

---

In addition, the straight-path interpolation achieves a better balance in converting a noisy image to its corresponding clean image compared to the reverse path as shown in Figure 4. This conversion is the shortest path in terms of Wasserstein distance (Moosmüller & Cloninger, 2020). In addition, this balanced conversion enables SPI-GAN learn a balanced mapping between latent vectors and images as well.

### 3.5 GENERATOR

We customize the generator architecture of StyleGAN2 (Karras et al., 2020b) for our purposes. Our generator takes $\mathbf{c}, \mathbf{s}$ as input and mimics the stochastic property of SGMs. However, the biggest difference from StyleGAN2 is that our generator is trained by the *continuous-time* mapping network which generates latent vectors at various interpolation points while maintaining one latent space across them. This is the key point in our model design to generate the interpolated image $\hat{\mathbf{i}}(u)$ with various $u$ settings.

### 3.6 DISCRIMINATOR

The discriminator of SPI-GAN is time-dependent, unlike the discriminator of traditional GANs. It takes $\hat{\mathbf{i}}(u)$ and the embedding of $u$ as input and learns to classify images from various interpolation points. As a result, it i) learns the data distribution which changes following the straight-path interpolation, and ii) solves the mode-collapse problem that traditional GANs have, since our discriminator sees various clean and noisy images. DD-GAN and Diffusion-GAN also use this strategy to overcome the mode-collapse problem. They give noise images following the reverse (forward) SDE path as input to the discriminator. As shown in Figure 4, however, the straight-path interpolation method maintains a better balance between noisy and clean images than the case where we sample following the SDE path. Therefore, our discriminator learns a more balanced set of noisy and clean images than that of DD-GAN and Diffusion-GAN. We refer to the Appendix for the detailed network structures.

### 3.7 TRAINING ALGORITHM

Our training algorithm is in Appendix Algorithm 2. In each iteration, we first create a mini-batch of $N$ real images, denoted $\{\mathbf{x}_0^l\}_{l=1}^N$. Using the forward SDE, we derive a mini-batch of $N$ noisy images, denoted $\{\mathbf{x}_T^l\}_{l=1}^N$. We then sample $u$, where $u \in (0, 1]$. After that, our mapping network generates a set of latent vectors, denoted $\{\mathbf{h}^l(u)\}_{l=1}^N$. Our generator then produces a set of fake images from the generated latent vectors. After that, we follow the standard adversarial training sequence. Our objective function to train our proposed model is in the Appendix.

### 3.8 HOW TO GENERATE

In order to generate samples with SPI-GAN, we need only $\mathbf{h}(1)$ from the mapping network. Unlike other auto-regressive denoising models that require multiple steps when generating samples, e.g., DD-GAN (cf. Figure 1 (b)), SPI-GAN learns the denoising path using a NODE-based mapping network. After sampling $\mathbf{z} \sim \mathcal{N}(\mathbf{0}, \sigma^2\mathbf{I})$, we feed them into the mapping network to derive $\mathbf{h}(1)$ — we solve the initial value problem in the Eq. 7 from 0 to 1 — and our generator generates a fake image $\hat{\mathbf{i}}(1)$. In other words, it is possible to generate latent vector $\mathbf{h}(1)$ directly in our case, which is later used to generate a fake sample $\hat{\mathbf{i}}(1)$ (sampling algorithm is in Algorithm 1).

Table 1: Results of the unconditional generation on CIFAR-10.

| Model | IS ↑ | FID ↓ | Recall↑ | NFE↓ |
|---|---|---|---|---|
| SPI-GAN (ours) | **10.2** | 3.01 | **0.66** | 1 |
| Diffusion-GAN (StyleGAN2) (Wang et al., 2022) | 9.94 | 3.19 | 0.58 | 1 |
| Denoising Diffusion GAN (DD-GAN), $K = 4$ (Xiao et al., 2021) | 9.63 | 3.75 | 0.57 | 4 |
| Score SDE (VP) (Song et al., 2021c) | 9.68 | 2.41 | 0.59 | 2000 |
| DDPM (Ho et al., 2020) | 9.46 | 3.21 | 0.57 | 1000 |
| NCSN (Song & Ermon, 2019) | 8.87 | 25.3 | - | 1000 |
| Adversarial DSM (Jolicoeur-Martineau et al., 2021b) | - | 6.10 | - | 1000 |
| Likelihood SDE (Song et al., 2021b) | - | 2.87 | - | - |
| Score SDE (VE) (Song et al., 2021c) | 9.89 | 2.20 | 0.59 | 2000 |
| Probability Flow (VP) (Song et al., 2021c) | 9.83 | 3.08 | 0.57 | 140 |
| LSGM (Vahdat et al., 2021) | 9.87 | **2.10** | 0.61 | 147 |
| DDIM, T=50 (Song et al., 2021a) | 8.78 | 4.67 | 0.53 | 50 |
| FastDDPM, T=50 (Kong & Ping, 2021) | 8.98 | 3.41 | 0.56 | 50 |
| Recovery EBM (Gao et al., 2021) | 8.30 | 9.58 | - | 180 |
| Improved DDPM (Nichol & Dhariwal, 2021) | - | 2.90 | - | 4000 |
| VDM (Kingma et al., 2021) | - | 4.00 | - | 1000 |
| UDM (Kim et al., 2021) | 10.1 | 2.33 | - | 2000 |
| D3PMs (Austin et al., 2021) | 8.56 | 7.34 | - | 1000 |
| Gotta Go Fast (Jolicoeur-Martineau et al., 2021a) | - | 2.44 | - | 180 |
| DDPM Distillation (Luhman & Luhman, 2021) | 8.36 | 9.36 | 0.51 | 1 |
| StyleGAN2 w/o ADA (Karras et al., 2020b) | 9.18 | 8.32 | 0.41 | 1 |
| StyleGAN2 w/ ADA (Karras et al., 2020a) | 9.83 | 2.92 | 0.49 | 1 |
| StyleGAN2 w/ Diffaug (Zhao et al., 2020) | 9.40 | 5.79 | 0.42 | 1 |

Table 2: Results on CelebA-HQ-256.

| MODEL | FID↓ |
|---|---|
| SPI-GAN (OURS) | **6.62** |
| DD-GAN | 7.64 |
| SCORE SDE | 7.23 |
| LSGM | 7.22 |
| UDM | 7.16 |
| PGGAN (KARRAS ET AL., 2018) | 8.03 |
| ADV. LA (PIDHORSKYI ET AL., 2020) | 19.2 |
| VQ-GAN (ESSER ET AL., 2021B) | 10.2 |
| DC-AE (PARMAR ET AL., 2021) | 15.8 |

Table 3: Results on LSUN-Church-256.

| MODEL | FID↓ |
|---|---|
| SPI-GAN (OURS) | 6.03 |
| DIFFUSION-GAN (STYLEGAN2) | 3.17 |
| DD-GAN | 5.25 |
| DDPM | 7.89 |
| IMAGEBART (ESSER ET AL., 2021A) | 7.32 |
| GOTTA GO FAST | 25.6 |
| PGGAN | 6.42 |
| STYLEGAN2 | 3.86 |
| CIPS (ANOKHIN ET AL., 2021) | **2.92** |

## 4 EXPERIMENTS

We describe our experimental environments and results. In the Appendix, more detailed experimental settings including software/hardware and hyperparameters, for reproducibility. We also release our model with trained checkpoints. More detailed experimental settings are in the Appendix.

### 4.1 EXPERIMENTAL ENVIRONMENTS

**Diffusion types.** Among various types, we conduct experiments based on the variance preserving SDE (VP-SDE) for their high sampling quality and reliability, which makes $\sigma^2 = 1$. That is, $\mathbf{i}(1)$ and $\mathbf{z}$ follow a unit Gaussian distribution (see Appendix for more descriptions).

**Datasets.** We use CIFAR-10 (Krizhevsky et al., 2014), CelebA-HQ-256 (Karras et al., 2018), and LSUN-Church-256 (Yu et al., 2015). CIFAR-10 has a resolution of 32x32 and is one of the most widely used datasets. CelebA-HQ-256 and LSUN-Church-256 contain high-resolution images of 256x256. Each of them has many real-world images.

**Evaluation metrics.** We use 5 evaluation metrics to quantitatively evaluate fake images. The inception score (Salimans et al., 2016) and the Fréchet inception distance (Heusel et al., 2017) are traditional methods to evaluate the fidelity of fake samples. The improved recall (Kynkäänniemi et al., 2019) reflects whether the variation of generated data matches the variation of training data. Finally, the number of function evaluations (NFE) and wall-clock time (Time) are used to evaluate the generation time for a batch size of 100 images.

### 4.2 MAIN RESULTS

In this subsection, we evaluate our proposed model quantitatively and qualitatively. For CIFAR-10, we perform the unconditional image generation task for fair comparisons with existing models. The quantitative evaluation results are shown in Table 1. Although our Fréchet inception distance (FID) is 0.6 worse than that of the Score SDE (VP), it shows better scores in all other metrics. However, our method has a better FID score than that of DD-GAN and Diffusion-GAN, which are the most

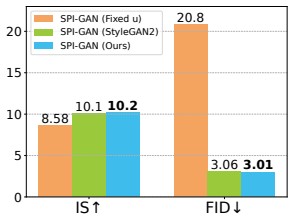 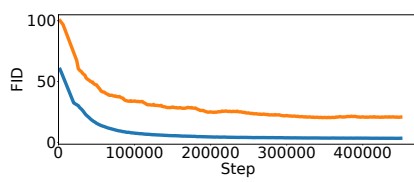

Figure 5: Qualitative results on CIFAR-10, CelebA-HQ-255, and LSUN-Church-256.

Figure 6: Ablation studies on CIFAR-10.

Figure 7: FID curves of the stochasticity with $u$ (blue) vs. the fixed $u$ (orange).

related methods. Diffusion-GAN, which follows the reverse SDE path using GAN, is inferior to our method for all those three quality metrics. LGSM also shows high quality for FID. However, its inception score (IS) and improved recall (Recall) scores are worse than ours.

Even for high-resolution images, our model shows good performance. In particular, our method shows the best FID score for CelebA-HQ-256 in Table 2, which shows the efficacy of our proposed method. However, our method does not produce significant improvements for LSUN-Church-256 in Table 3 — our method outperforms DD-GAN with more improved metrics in Table 4. The qualitative results are in Figure 5. As shown, our method is able to generate visually high-quality images. More detailed images are in the Appendix.

## 4.3 ABLATION STUDIES

In this subsection, we conduct experiments by i) fixing the intermediate points, and ii) changing the mapping network, which are the two most important.

**Fixed intermediate points.** In our model, $u$ is randomly sampled between 0 and 1 to learn a latent vector's path. However, we can fix $u = \frac{1}{2}$. That is, it does not learn about various $u$, but only trains with a specific section. It can be seen that in Figure 6 and Figure 7, however, our original setting not only shows better sampling quality than the fixed setting but also converges more rapidly.

**Continuous-time mapping network.** Our NODE-based mapping network learns the straight-path between $x_0$ and $x_1$ along $u$. In order to see the effect of our proposed continuous-time mapping network, we applied the mapping network proposed by StyleGAN2 to our model. As a result, Figure 6 shows that the continuous-time mapping network that we proposed shows good overall performance. In Appendix C, we show that our NODE-based mapping network is able learn appropriate denoising process whereas the original mapping network of StyleGAN2 is not, which gives us more insights that Figure 6.

## 4.4 ADDITIONAL STUDIES

We introduce additional studies evaluating sampling quality, sampling time, and interpolations.

**Improved metrics.** FID is one of the most popular evaluation metrics to measure the similarity between real and generated images. Generated images are sometimes evaluated for fidelity and diversity using improved precision and recall (Kynkäänniemi et al., 2019). However, the recall does not accurately detect similarities between

Table 4: Improved quality metrics on LSUN-Church-256.

| Model | Recall↑ | Coverage↑ |
|---|---|---|
| SPI-GAN (Ours) | 0.28 | **0.65** |
| Diffusion-GAN (StyleGAN2) | 0.16 | 0.38 |
| DD-GAN | 0.16 | 0.58 |
| CIPs | **0.43** | 0.57 |

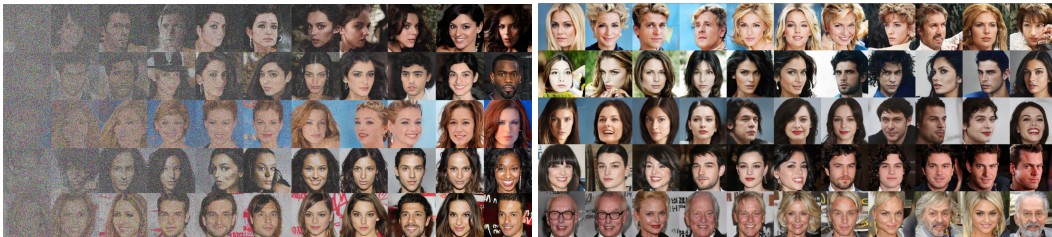

Figure 8: **Left:** Generation by varying the latent vector from $\mathbf{h}(0)$ to $\mathbf{h}(1)$ given fixed $\mathbf{z}$. **Right:** Generation by interpolating $\mathbf{z} = (1-a)\mathbf{z}' + a\mathbf{z}''$, where $0 \leq a \leq 1$.

two distributions and is not robust against outliers.

Therefore, we evaluate the generated images with the coverage (Naeem et al., 2020) which overcomes the limitations of the recall. We compare our method with CIPs, which marks the best FID score in LSUN-Church-256, and the results are in Table 4. Our SPI-GAN outperforms CIPs in terms of coverage.

**Sampling time analyses.** Our model shows outstanding performance in all evaluation metrics compared to DD-GAN and Diffusion-GAN. In particular, SPI-GAN, unlike DD-GAN, does not increase the sample generation time — in fact, our method only affects the training time because we train our method with $u \in (0, 1]$. However, we always use $u = 1$ for generating a clean image $\hat{\mathbf{i}}(1)$.

Table 5: The Generation time comparison

| Model | Time |
|---|---|
| SPI-GAN (Ours) | 0.04 |
| Diffusion-GAN (StyleGAN2) | 0.04 |
| DD-GAN | 0.36 |
| StyleGAN2 | 0.04 |

Therefore, our method is fast in generating images after being trained, which is one good characteristic of our method. We measure the wall-clock runtime 10 times for CIFAR-10 with a batch size of 100 using an A5000 GPU to evaluate the sampling time. We applied the same environment for fair comparisons. As a result, our method's sample generation time in Table 5 is almost the same as that of StyleGAN2, which is one of the fastest methods. In summary, SPI-GAN not only increases the quality of samples but also decreases the sampling time.

**Generation by manipulating z and h.** There are three manipulation parts in our model. The first one is generated by changing the latent vector from $\mathbf{h}(0)$ to $\mathbf{h}(1)$, the second one is the interpolation between two noisy vectors $\mathbf{z}'$ and $\mathbf{z}''$ in Figure 8, and the last one is the interpolation between two latent vectors $\mathbf{h}(1)'$ to $\mathbf{h}(1)''$ in Appendix Figure 11. In Figure 8, the ideal generation is that noises are gradually removed for an image, but similar images, not the same, are produced. However, the denoising patterns can be observed well. In addition, Figure 8 shows the interpolation of the noise vector ($\mathbf{z}$). Interpolation of the latent vector ($\mathbf{h}(1)$) is in Appendix Figure 11. One can observe that generated images are gradually changed from a mode to another.

## 5 CONCLUSIONS AND DISCUSSIONS

Score-based generative models (SGMs) now show the state-of-the-art performance in image generation. However, the sampling time of SGMs is significantly longer than other generative models, such as GANs, VAEs, and so on. Therefore, we presented the most balanced model by reducing the sampling time, called SPI-GAN. Our method is a GAN-based approach that imitates the straight-path interpolation. The straight denoising path is the most optimal path in terms of the Wasserstein distance and is simple enough for the model to be easy to learn. Moreover, it can directly generate fake samples without any recursive computation (or step). Our method shows the best sampling quality in various metrics and faster sampling time than other score-based methods. Our ablation and additional studies show the effectiveness of our proposed model. One limitation is that our method fails to achieve the best results for FID in LSUN-Church-256. However, our method's coverage is the highest for it. All in all, one can see that SPI-GAN is one of the most balanced methods among the generative task trilemma's criteria: sampling quality, diversity, and time.

## 6 ETHICS STATEMENT

Generative models are growing rapidly. In particular, score-based generative models generate more realistic images. In this situation, our proposed model can generate high-quality images quickly by reducing inference time significantly compared to previous models. Although there are positive aspects to this research direction, there may be negative aspects such as malicious video generation and image synthesis. Generative models are growing rapidly. In particular, score-based generative models generate more realistic images. In this situation, our proposed model can generate high-quality images quickly by reducing inference time significantly compared to previous models. Although there are positive aspects to this research direction, there may be negative aspects such as malicious video generation and image synthesis.

## 7 REPRODUCIBILITY STATEMENT

For reproducibility, we attached the sources codes and trained checkpoints in our supplementary materials. There are detailed descriptions for experimental environment settings, datasets, training processes, evaluation and visualization processes in README.md. In Appendix, we also list all the detailed neural network architectures and their hyperparameters.

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

APPENDIX

## A    STOCHASTIC DIFFERENTIAL EQUATION (SDE)

In (Song et al., 2021c), three types of SDE diffusion processes are presented. Depending on the type, $f(\mathbf{x}, t)$ and $g(t)$ are defined as follows:

$$f(\mathbf{x}, t) = \begin{cases} 0, & \text{if VE-SDE,} \\ -\frac{1}{2}\beta(t)\mathbf{x}, & \text{if VP-SDE,} \\ -\frac{1}{2}\beta(t)\mathbf{x}, & \text{if sub-VP-SDE,} \end{cases} \tag{8}$$

$$g(t) = \begin{cases} \sqrt{\frac{d[\sigma^2(t)]}{dt}}, & \text{if VE-SDE,} \\ \sqrt{\beta(t)}, & \text{if VP-SDE,} \\ \sqrt{\beta(t)(1 - e^{-2\int_0^t \beta(s)\,ds})}, & \text{if sub-VP-SDE,} \end{cases} \tag{9}$$

where $\sigma^2(t)$ and $\beta(t)$ are functions w.r.t. time $t$. Full derivatives of VE, VP and sub-VP SDE are presented in (Song et al., 2021c, Appendix. B).

## B    EFFECTIVENESS OF THE STRAIGHT-PATH INTERPOLATION

Between $\mathbf{x}_T$ and $\mathbf{x}_0$, our straight-path interpolation provides a much simpler path than that of the SDE path because it follows the linear equation in Eq. equation 6. Also, because of the nature of the linear interpolation, its training is robust, even when $\mathbf{i}(u + \Delta u)$ is missing, if $\mathbf{i}(u)$ and $\mathbf{i}(u + 2\Delta u)$ are considered. This is not guaranteed if a path from $\mathbf{x}_T$ to $\mathbf{x}_0$ is non-linear. As a result, SPI-GAN using the straight-path interpolation shows better performance.

One can also derive the following ordinary differential equation (ODE) from the straight-path interpolation definition in Eq. equation 6 after taking the derivative w.r.t. $u$:

$$\frac{d\mathbf{i}(u)}{du} = \mathbf{x}_0 - \mathbf{x}_T, \tag{10}$$

where one can get $\mathbf{i}(u + h) = \mathbf{i}(u) + h\frac{d\mathbf{i}(u)}{du} = (u + h)\mathbf{x}_0 + (1 - u - h)\mathbf{x}_T$ with the Euler method. In comparison with the SDE in Eqs. equation 8 and equation 9, the ODE provides a much simpler process.

## C    EFFECTIVENESS OF NEURAL ORDINARY DIFFERENTIAL EQUATIONS-BASED MAPPING NETWORK

As we mentioned in the related work section, NODEs are able to model continuous dynamics of hidden vectors over time using the following method:

$$\mathbf{h}(u) = \mathbf{h}(0) + \int_0^u f(\mathbf{h}(t), t; \boldsymbol{\theta_f})dt, \tag{11}$$

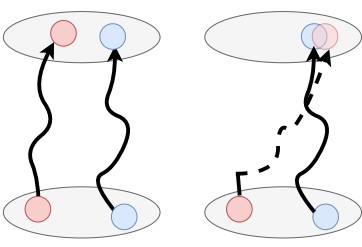

Homeomorphic          Non-Homeomorphic

where the neural network $f(\mathbf{h}(t), t; \boldsymbol{\theta_f})$ learns $\frac{d\mathbf{h}(t)}{dt}$. To derive $\mathbf{h}(u)$, we solve the integral problem, and in this process, there is one well-known characteristic of NODEs. Let $\psi_t : \mathbb{R}^{\dim(\mathbf{h}(0))} \to \mathbb{R}^{\dim(\mathbf{h}(u))}$ be a mapping from 0 to $u$ generated by an ODE after solving the integral problem. It is widely known that $\psi_t$ becomes a homeomorphic mapping: $\phi_t$ is continuous and bijective and $\phi_t^{-1}$ is also continuous for all $t \in [0, T]$, where $T$ is the last time point of the time domain (Dupont et al., 2019; Massaroli et al., 2020). From this characteristic, the following proposition can be derived: the topology of the input space of $\phi_t$ is

Figure 9: In a homeomorphic mapping, the relative positions of the red and blue particles cannot be switched after the mapping.

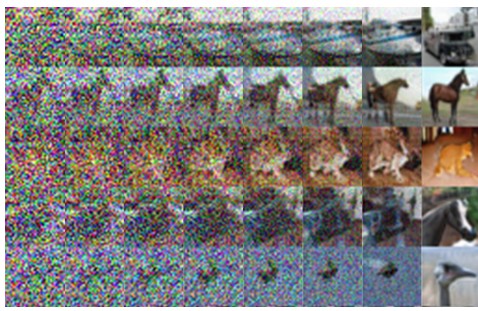 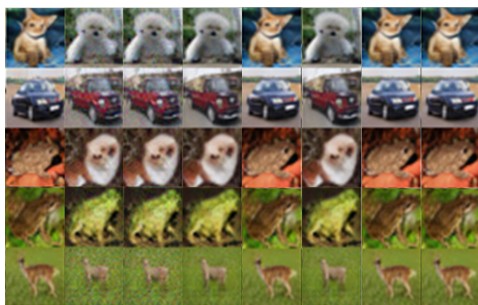

Figure 10: Comparing $\hat{\mathbf{i}}(u)$ according to the mapping network type. Given a fixed z, $\hat{\mathbf{i}}(u)$ generated by varying the latent vector from $\mathbf{h}(0)$ to $\mathbf{h}(1)$ **Left:** our NODE-based mapping network. **Right:** the StyleGAN2's original mapping network which is quickly trained to overlook $u$

preserved in its output space, and therefore, hidden trajectories crossing each other cannot be represented by NODEs — one can consider that topology as relative positions among particles. Therefore, our NODE-based mapping network can learn the hidden dynamics of $\mathbf{h}(u)$ for all $u \in (0, 1]$ while maintaining the topology of $\mathbf{h}(u)$ at $t = 0$ (e.g., Fig. 9).

Figure 10 shows one advantage of the homeomorphic mapping of SPI-GAN. SPI-GAN with a NODE-based mapping network, shows an appropriate denoising process when $\hat{\mathbf{i}}(0)$ is generated using $\mathbf{h}(0)$ to $\mathbf{h}(1)$, given a fixed $z$. In contrast, StyleGAN2's mapping network (non-homeomorphic) do not show a denoising process. In other words, NODE-based SPI-GAN with the homeomorphic characteristic can learn the denoising process, but SPI-GAN with the non-homeomorphic mapping network are collapsed into clean images only. We think that this is because i) the time information $u$ is concatenated with the input to the StyleGAN2's original non-homeomorphic mapping network, but ii) it is trained to overlook $u$. This justifies our design choice to explicitly model the hidden dynamics with the homeomorphic NODE.

# D  SPI-GAN DETAILS

In this section, we refer to the detailed model architecture, object function, and training algorithm of our proposed SPI-GAN.

## D.1  MODEL ARCHITECTURE

The network architectures of StyleGAN2 are modified to implement our proposed straight-path interpolation after adding the NODE-based mapping network and customizing some parts.

**Mapping network.** Our mapping network consists of two parts. First, the network architecture to define the function $\mathbf{o}$ is in Table 6. Second, the NODE-based network has the following ODE function $\mathbf{r}$ in Table 7.

Table 6: The architecture of the network $\mathbf{o}$.

| LAYER | DESIGN | INPUT SIZE | OUTPUT SIZE |
|---|---|---|---|
| 1 | LEAKYRELU(LINEAR) | $C \times H \times W$ | $\dim(\mathbf{h})$ |

Table 7: The architecture of the network $\mathbf{r}$.

| LAYER | DESIGN | INPUT SIZE | OUTPUT SIZE |
|---|---|---|---|
| 1 | LEAKYRELU(LINEAR) | $\dim(\mathbf{h})$ | $\dim(\mathbf{h})$ |

**Generator.** We follow the original StyleGAN2 architecture. However, we use the latent vector $\mathbf{h}(u)$ instead of the intermediate latent code w of StyleGAN2.

**Discriminator.** The network architecture of the discriminator is also based on StyleGAN2 (Karras et al., 2020b) — StyleGAN2 has two versions for the discriminator, i.e., Original and Residual. However, our discriminator receives time $u$ as a conditional input. To this end, we use the positional embedding of the time value as in (Ho et al., 2020). The hyperparameters for the discriminator are in Table 8.

## D.2 OBJECT FUNCTION

We train our model using the Adam optimizer for training both the generator and the discriminator. We use the exponential moving average (EMA) when training the generator, which achieves high performance in (Ho et al., 2020; Song et al., 2021c; Karras et al., 2020a). The hyperparameters for the optimizer are in Table 8. The adversarial training object of our model is as follows:

$$\min_{\phi} \mathbb{E}_{\mathbf{i}(u) \sim \mathbf{q}_{\mathbf{i}(u)}} \big[ - \log(D_{\phi}(\mathbf{i}(u), u))$$
$$+ \mathbb{E}_{\mathbf{z} \sim \mathcal{N}(\mathbf{0}, \sigma^2 \mathbf{I})} \big[ - \log(1 - D_{\phi}(G_{\theta}(M_{\psi}(\mathbf{z})), u)) \big] \big], \quad (12)$$
$$\max_{\theta, \psi} \mathbb{E}_{\mathbf{i}(t) \sim \mathbf{q}_{\mathbf{i}(u)}} \big[ \mathbb{E}_{\mathbf{z} \sim \mathcal{N}(\mathbf{0}, \sigma^2 \mathbf{I})} \big[ \log(D_{\phi}(G_{\theta}(M_{\psi}(\mathbf{z})), u)) \big] \big],$$

where, $\mathbf{q}_{\mathbf{i}(u)}$ is the interpolated image distribution, $D_{\phi}$ is denoted as the discriminator, $G_{\theta}$ is denoted as the generator, and $M_{\psi}$ is denoted as the mapping network of our model. We also use the $R_1$ regularization and the path length regularization (Karras et al., 2020b). $\lambda_{R_1}$ (resp. $\lambda_{path}$) means the coefficient of the $R_1$ regularization term (resp. the coefficient of the path length regularization term). Each regularizer term is as follows:

$$R_1(\phi) = \lambda_{R_1} \mathbb{E}_{q(\mathbf{i}(u))} \big[ \| \nabla_{\mathbf{i}(t)} (D_{\phi}(\mathbf{i}(u)|u)) \|^2 \big], \quad (13)$$
$$\text{Path length} = \lambda_{path} \mathbb{E}_{\mathbf{h}(u), \mathbf{i}(u)} \big( \| \mathbf{J}_{\mathbf{h}(u)}^T \mathbf{i}(u) \|_2 - a \big), \quad (14)$$

where $\mathbf{J}_{\mathbf{h}(u)} = \partial G_{\theta}(\mathbf{h}(u)) / \partial \mathbf{h}(u)$ is the Jacobian matrix. The constant $a$ is set dynamically during optimization to find an appropriate global scale. The path length regularization helps with the mapping from latent vectors to images. The lazy regularization makes training stable by computing the regularization terms ($R_1$, path length) less frequently than the main loss function. In SPI-GAN, the regularization term for the generator and the discriminator is calculated once every 4 iterations and once every 16 iterations, respectively. The hyperparameters for the regularizers are in Table 8.

## D.3 TRAINING ALGORITHM

There is a training algorithm. The two main differences with the sampling algorithm are i) we sample a set of noisy vectors $\{\mathbf{z}^l\}_{l=1}^N$ whereas we use $\{\mathbf{x}_T^l\}_{l=1}^N$ in the training algorithm, and ii) for sampling, we need only $\{\mathbf{h}^l(1)\}_{l=1}^N$.

---

**Algorithm 2** How to train SPI-GAN

**Input**: Training data $D_{\text{train}}$, Maximum iteration numbers $max\_iter$

1: Initialize discriminator $\phi$, mapping net. $\psi$, generator $\theta$
2: $iter \leftarrow 0$
3: **while** $iter < max\_iter$ **do**
4:     Create a mini-batch of real images $\{\mathbf{x}_0^l\}_{l=1}^N$, where $\mathbf{x}_0^l$ means $l$-th real image
5:     Calculate a mini-batch of noisy images $\{\mathbf{x}_T^l\}_{l=1}^N$ with the forward SDE path
6:     Sample $u$, where $u \in (0, 1]$
7:     Calculate $\{\mathbf{h}^l(u)\}_{l=1}^N$ with the mapping network which processes $\{\mathbf{x}_T^l\}_{l=1}^N$
8:     Generate fake images $\{\hat{\mathbf{i}}^l(u)\}_{l=1}^N$ with the generator
9:     **if** $iter \mod 2 \equiv 0$ **then**
10:         Calculate $\{\mathbf{i}^l(u)\}_{l=1}^N$ with Eq. 6
11:         Update $\phi$ via adversarial training
12:     **else**
13:         Update $\psi$ and $\theta$ via adversarial training
14:     **end if**
15:     $iter \leftarrow iter + 1$
16: **end while**
17: **return** $\phi, \psi, \theta$

---

## E EXPERIMENTAL DETAILS

In this section, we describe the detailed experimental environments of SPI-GAN. We build our experiments on top of (Kang et al., 2022)

### E.1 EXPERIMENTAL ENVIRONMENTS

Our software and hardware environments are as follows: UBUNTU 18.04 LTS, PYTHON 3.9.7, PYTORCH 1.10.0, CUDA 11.1, NVIDIA Driver 417.22, i9 CPU, NVIDIA RTX A5000, and NVIDIA RTX A6000.

### E.2 TARGET DIFFUSION MODEL

Our model uses a forward SDE to transform an image ($\mathbf{x}_0$) into a noise vector ($\mathbf{x}_T$). When generating a noise vector, we use the forward equation of VP-SDE for its high efficacy/effectiveness. The $\beta(t)$ function of VP-SDE is as follows:

$$\beta(t) = \beta_{\min} + t(\beta_{\max} - \beta_{\min}), \tag{15}$$

where $\beta_{\max}$ = 20, $\beta_{\min}$ = 0.1, and $t' := \frac{t}{T}$ which is normalized from $t \in \{0, 1, \ldots, T\}$ to $[0, 1]$. Under these conditions, (Song et al., 2021c, Appendix B) proves that the noise vector at $t' = 1$ ($\mathbf{x}_T$) follows a unit Gaussian distribution.

### E.3 DATA AUGMENTATION

Our model uses the adaptive discriminator augmentation (ADA) (Karras et al., 2020a), which has shown good performance in StyleGAN2.[2] The ADA applies image augmentation adaptively to training the discriminator. We can determine the maximum degree of the data augmentation, which is known as an ADA target, and the number of the ADA learning can be determined through the ADA interval. We also apply mixing regularization ($\lambda_{mixing}$) to encourage the styles to localize. Mixing regularization determines how many percent of the generated images are generated from two noisy images during training (a.k.a, style mixing). There are hyperparameters for the data augmentation in Table 8.

### E.4 HYPERPARAMETERS

We list all the key hyperparameters in our experiments for each dataset. Our supplementary material accompanies some trained checkpoints and one can easily reproduce.

Table 8: Hyperparameters set for SPI-GAN.

|  |  | CIFAR-10 | CelebA-HQ-256 | LSUN-Church-256 |
|---|---|---|---|---|
| | ADA target | 0.6 | 0.6 | 0.6 |
| Augmentation | ADA interval | 4 | 4 | 4 |
| | $\lambda_{mixing}$ (%) | 0 | 90 | 90 |
| Architecture | Mapping network | 1 | 7 | 7 |
| | Discriminator | Original | Residual | Residual |
| | Learning rate for generator | 0.0025 | 0.0025 | 0.0025 |
| Optimizer | Learning rate for discriminator | 0.0025 | 0.0025 | 0.0025 |
| | EMA | 0.999 | 0.999 | 0.999 |
| | ODE Solver | | 4th order Runge–Kutta | |
| | Lazy generator | 4 | 4 | 4 |
| Regularization | Lazy discriminator | 16 | 16 | 16 |
| | $\lambda_{R_1}$ | 0.01 | 10 | 10 |
| | $\lambda_{path}$ | 0 | 2 | 2 |

### E.5 TRAINING TIME

The training time for each 1024 CIFAR-10 images is around 32.0s for SPI-GAN and around 45.6s for Diffusion-GAN using four NVIDIA A5000 GPUs.

### E.6 ADDITIONAL ABLATION STUDIES

We report additional ablation studies to show the superiority of our model. First, the result of SPI-GAN without a mapping network is reported in Table 9. To generate a noise image $\hat{\mathbf{i}}(u)$ without

---

[2]https://github.com/NVlabs/stylegan2 (Nvidia Source Code License)

a mapping network, $u$ is given as a conditional input to the generator. Second, give time point $u$ as a conditional input to the SPI-GAN generator. The result is in Table 10. SPI-GAN shows better performance than both ablation studies.

Table 9: Ablation study for mapping network

| MODEL | FID |
|---|---|
| SPI-GAN (W/O MAPPING NETWORK) | 5.72 |
| SPI-GAN | 3.01 |

Table 10: Ablation study for condition $u$

| LAYER | DESIGN |
|---|---|
| SPI-GAN (CONDITION $u$ TO GENERATOR) | 3.03 |
| SPI-GAN | 3.01 |

## F  VISUALIZATION

We introduce interpolation and several high-resolution generated samples.

### F.1  INTERPOLATION

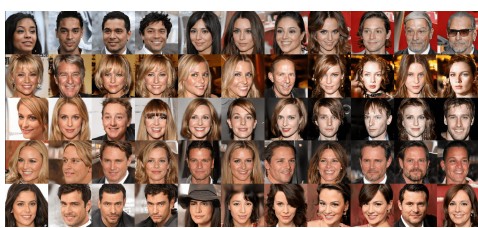

Figure 11: Generation by interpolating $\mathbf{h}(1) = (1-a)\mathbf{h}(1)' + a\mathbf{h}(1)''$, where $0 \leq a \leq 1$.

### F.2  CIFAR-10

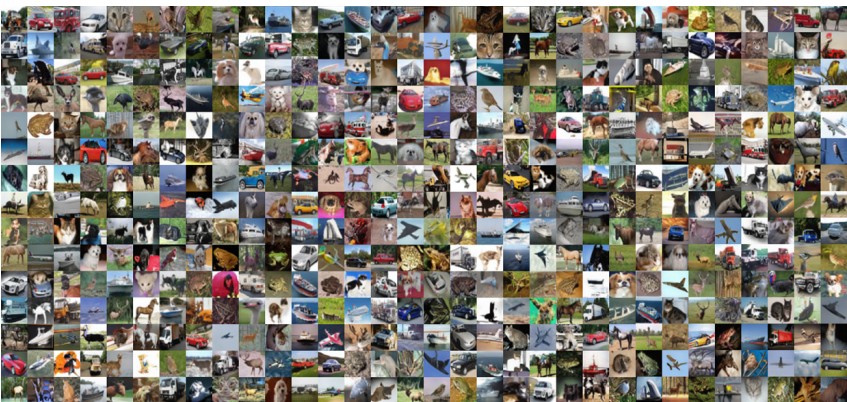

Figure 12: Qualitative results on CIFAR-10.

### F.3 QUALITATIVE RESULTS ON HIGH RESOLUTION DATASETS

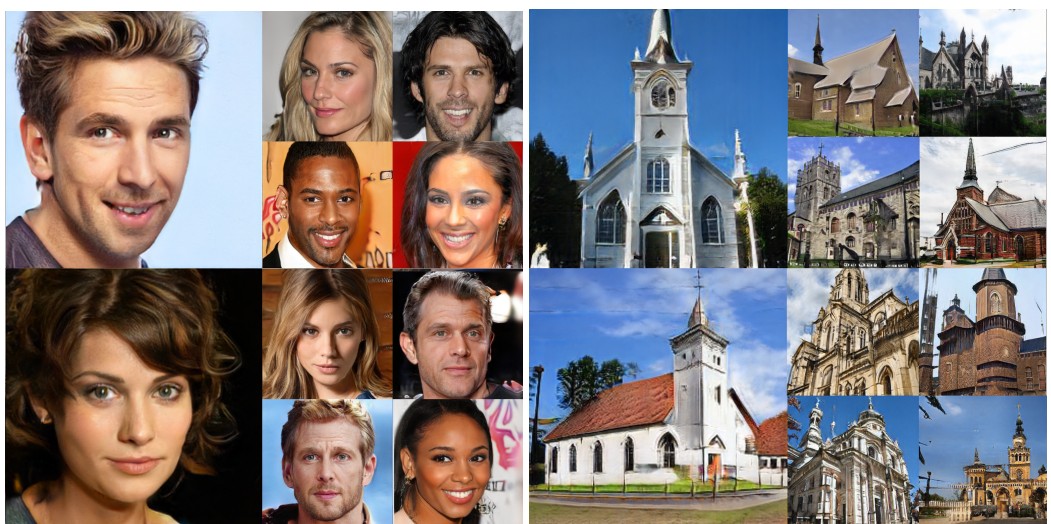

Figure 13: CelebA-HQ-256

Figure 14: LSUN-Church-256

### F.4 COMPARISON OF GENERATED DIFFUSION PROCESSES AND REAL DIFFUSION PROCESSES

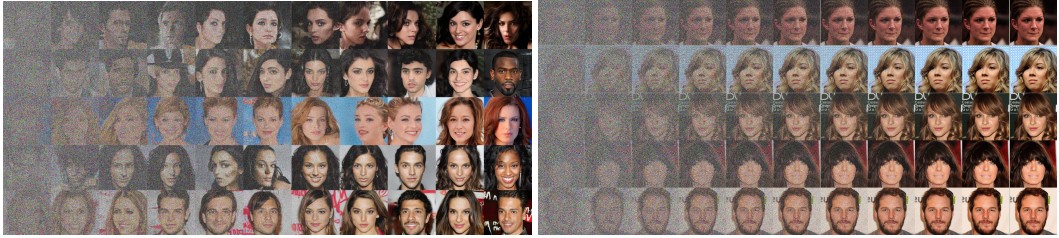

Figure 15: Comparing $\hat{\mathbf{i}}(u)$ and $\mathbf{i}(u)$. **Left:** Given a fixed $\mathbf{z}$, $\hat{\mathbf{i}}(u)$ is generated from SPI-GAN by varying the latent vector from $\mathbf{h}(0)$ to $\mathbf{h}(1)$. **Right:** Diffusion process of original images

