# OpenReview forum: "SPI-GAN: Denoising Diffusion GANs with Straight-Path Interpolations"
_ICLR.cc/2024/Conference — Submitted to ICLR 2024_

### Official Review · Reviewer_NZnM · 2023-10-29

**Soundness:** 3 good
**Presentation:** 3 good
**Contribution:** 3 good
**Rating:** 5
**Confidence:** 3

**Summary:**

The paper proposes SPI-GAN, a GAN-based denoising diffusion model that provides the best balance among the sampling quality, diversity, and time. The key idea is to predict the straight-path interpolated image i(u) of the target time-step u. SPI-GAN includes a mapping module, a generator, and a discriminator. The mapping module consists of an initial embedding network o to encode the random noise input (i(0)) and a NODEs-based mapping network h that converts that initial embedding to the denoised embedding at the target time-step u. The generator inputs that denoised embedding and predicts the straight-path interpolated image, while the discriminator differentiates between real and predicted i(u). SPI-GAN employs many components of StyleGAN2, including the generator and ADA augmentation. SPI-GAN can perform 1-step diffusion to achieve real-time speed while outperforming other GAN+Diffusion approaches in most evaluation scores.

**Strengths:**

- SPI-GAN can perform 1-step diffusion to achieve real-time speed, which is almost similar to StyleGAN2.
- Even with 1-step diffusion, SPI-GAN can achieve quite good performance. On CIFAR-10, it obtains the best Recall and outperforms other GAN+Diffusion approaches in FID. On CelebA-HQ-256, SPI-GAN acquires a reasonable FID score, which outperforms other GAN+Diffusion methods and some score-based model representatives. On LSUNChurch-256, it obtains a high coverage, outperforming CIPS and the GAN+Diffusion counterparts.
- The writing is clear and easy to follow.

**Weaknesses:**

- SPI-GAN is highly grounded by StyleGAN2 (generator's backbone, ADA training), and I am not sure if the score-based components are important.
From Fig.6, by replacing the NODEs-based mapping network with StyleGAN2's mapping network, the model's performance is almost unchanged. That new network is pretty much StyleGAN2-Ada, but instead of modeling the clean image only, the model learns to generate the linear-interpolated version between the clean image (i(1)) and its full-noise version (i(0)), conditioning on an interpolation factor u. The NODEs-based mapping network, while being sophisticated, may not be necessary and can be replaced by a simple MLP-based solution. It makes SPI-GAN look like a pure GAN model, disguising itself as a diffusion model. It does not devalue the paper but changes its narrative.

- In Fig.8-Left, given the same initial embedding, latent vectors h(u) at different time-step u produce completely different output images. It does not match the expected behavior of diffusion models.

- Can SPI-GAN perform multi-step (NFE > 1), and if so, does the performance improve?

- The authors should ablate the model performance when adding the time-step (u) input to the generator. Ablation studies on the role of s are also recommended.

- SPI-GAN has a worse FID score compared with other Diffusion+GAN counterparts on LSUN-Church.

- Fig 7: The x ticker labels are too close to each other. The caption should be improved, e.g., "stochastic" -> "stochasticity".

**Questions:**

- Can you replace the NODEs-based mapping network with StyleGAN2's mapping network, and report the performance of that model in Table 1-3? Does SPI-GAN have noticable better performance compared with that pure GAN method?
- In Fig.8-Left, given the same initial embedding, latent vectors h(u) at different time-step u produce completely different output image. It does not match the expected behavior of diffusion models. Can you explain why?
- Can SPI-GAN perform multi-step (NFE > 1), and if so, does the performance improve?
- The authors should ablate the model performance when adding the time-step (u) input to the generator. Ablation studies on the role of s is also recommended.

---

> ### Author Response · Authors · 2023-11-21
>
> Thank you for the encouraging remarks about our paper’s contribution and presentation, and the valuable feedback from the reviewer. We hope that our responses will solve the reviewer’ questions.
>
> > Can you replace the NODEs-based mapping network with StyleGAN2's mapping network, and report the performance of that model in Table 1-3? Does SPI-GAN have noticeable better performance compared with that pure GAN method?
>
> $\to$ In Fig.6, there is a slight degradation in performance even after replacing the NODE-based mapping network with the StyleGAN2's original mapping network. The StyleGAN2's original mapping network is able to produce good images because the condition $u$ is given to it. In other words, we concatenate the time condition $u$ with input to the StyleGAN2's mapping network.
> However, there is a difference between our NODE-based mapping network and the StyleGAN2’s mapping network. NODE-based mapping networks have homeomorphic characteristics. Because of this homomorphic characteristic, SPI-GAN with a NODE-based mapping network shows an appropriate denoising process when $\hat{\mathbf{i}}(0)$ is generated using $\mathbf{h}(0)$ to $\mathbf{h}(1)$, given a fixed $z$. In contrast, StyleGAN2's mapping network, which has non-homeomorphic properties, does not show a denoising process even when time point $u$ is used as a condition. A visualization of the difference between the NODE-based mapping network and the StyleGAN2 mapping network  is in Figure 10 in Appendix C in our revised paper.
>
> > In Fig.8-Left, given the same initial embedding, latent vectors h(u) at different time-step u produce completely different output image. It does not match the expected behavior of diffusion models. Can you explain why?
>
> $\to$ SPI-GAN learns the data distribution at arbitrary time $u \in [0,1]$. The diffusion model has an auto-regressive denoising process and therefore, one sample is continuously denoised. However, SPI-GAN does not do it but it can learn the ground-truth data distribution at time $u$.
>
> > Can SPI-GAN perform multi-step (NFE > 1), and if so, does the performance improve?
>
> $\to$ The multi-step generation is not needed since our method is not auto-regressive but learns the data distribution at time $u$. With this property, SPI-GAN can generate images $\hat{\mathbf{i}}(1)$ directly without querying intermediate timepoints. For training purposes, however, it is helpful to train with all $u \in [0,1]$ since it augments training data with continuously evolving data distributions over time.
>
> > The authors should ablate the model performance when adding the time-step (u) input to the generator. Ablation studies on the role of s is also recommended.
>
> $\to$ Thank you for the suggestion. As your suggestion, we report the results of ablate the model performance when adding the time-step ($u$) input to the generator in the following table. Table shows that there is no significant difference from SPI-GAN even if the time point $u$ is explicitly given to the generator as a condition. This can be confirmed in Table 10 in Appendix E.6 in the revision. However, $s$ used as input to the generator is a layer-wise varying noise, which was proposed in StyleGAN2 for stochasticity and is not the contribution of our SPI-GAN. After removing $s$, the stochasticity of our model is removed, which is not appropriate when the original diffusion model is stochastic.
>
> |            CIFAR-10           |  FID |
> |:-----------------------------:|:----:|
> | SPI-GAN (condition $u$ to generator) | 3.03 |
> |            SPI-GAN            | 3.01 |
>
> > Fig 7: The x ticker labels are too close to each other. The caption should be improved, e.g., "stochastic" -> "stochasticity".
>
> $\to$ Thank you for your comments. We revised Figure 7 and its caption in the revised paper for clarity.

---

> ### Author Response · Authors · 2023-11-22
>
> Dear reviewer NZnM.
>
> We apologize for the late author response during the author/reviewer discussions period. We did our best to conduct additional experiments to help reviewers understand.
>
> Please leave more questions if any. We are ready to answer all your potential unclear points to help your understanding.
>
> Best regards, Authors

---

> ### Author Response · Authors · 2023-11-23
>
> Dear reviewer NZnM
>
> We appreciate the reviewer’s time and effort in reviewing our manuscript and insightful comments.
>
> As the closure of the discussion period is approaching, we would like to bring the review’s attention and check if the reviewer could let us know whether the concerns or the misunderstanding have been addressed by our response.
>
> If this is the case, we would appreciate if you could adjust your rating accordingly.
>
> Best regards, Authors

---

> ### Comment · Reviewer_NZnM · 2023-12-01
> **Thanks for the rebuttal. I decided to keep my initial evaluation.**
>
> Many thanks for your answers. Unfortunelatelt, this rebuttal did not adequately address some of my concerns.
> - First, the authors did not provide the performance of SPI-GAN with the StyleGAN2 mapping network for comparison in Table 1-3.
> - Instead, they showed that the NODE-based mapping network allowed to produce an appropriate denoising process, while the StyleGAN2 mapping network could not, as shown in Fig.10. While the figure looks good, it does not directly solve my concern and does not change the fact that the NODE-based mapping network only provides a marginal improvement compared with the StyleGAN2 one. Adding u as a condition to the StyleGAN2 mapping network also does not make sense to me; it is expected to produce different output images. Finally, I am quite confused about how SPI-GAN can produce consistent denoising in Fig. 10 but very inconsistent ones in Fig.8-Left.
> - The answer on the inconsistency in Fig.8-Left is not clear and convincing.
>
> Besides, many concerns raised by vo3V are critical but not properly addressed.
>
> Based on these points, I decided to keep my initial evaluation.
>
> Sincerely,
>
> Reviewer NZnM

---

### Official Review · Reviewer_vo3V · 2023-10-31

**Soundness:** 1 poor
**Presentation:** 3 good
**Contribution:** 2 fair
**Rating:** 5
**Confidence:** 4

**Summary:**

This paper proposes solving the generative trilemma by using a GAN approach that imitates the denoising process of a score-based diffusion model. The method uses a time-conditioned generator (implicitly by the latent) and discriminator. The denoising process is linear, unlike SDE-based diffusion, and the latent is computed by a neural ODE.

**Strengths:**

- The authors have succeeded in presenting their research in a clear way, and the background work is relevant.
- The integration of optimal transport ideas into the context of diffusion is a novel and intriguing concept. This innovative approach not only contributes to the paper's originality but also has the potential to inspire further research in this promising direction.
- The paper's method seems to achieve a shorter interpolation path between images and noise. This outcome opens up new opportunities for further study of improved noise schedulers in diffusion model theory.
- The introduction of the general idea of straight-path interpolation deviates from traditional diffusion trajectories and introduces a new perspective on how denoising can be easily handled. This again could lead to further research.

**Weaknesses:**

- The NODE map, as evidenced in Figure 6, appears to offer marginal improvements over the vanilla mapping network from StyleGAN2. Consequently, the novelty of the method diminishes, as it seems to reduce to an image and time-conditioned StyleGAN. The authors should address how the method distinguishes itself more significantly from existing approaches.
- In a quantitative image quality comparison, the proposed method does not appear to clearly outperform vanilla StyleGAN2 in Tables 1-3. For instance, StyleGAN2 trained with ADA (same as SPI-GAN as shown in Sup. Mat. Table 8) exhibits superior performance in Table 1. Additionally, in Table 2, a StyleGAN2 comparison is missing, but running the official StyleGAN2 implementation on CelebA-HQ-256 with default settings (gamma=1, noaug) yields an FID score of 5, lower than the proposed method. A comprehensive quantitative analysis is required to establish the method's advantage.
- The claim of achieving diversity as good as score-based models is made based on datasets that are, in some cases, insufficient for assessing diversity. The resolution of the datasets is often low, and they are primarily single-domain (faces and churches). To validate the diversity claim, it is essential to perform experiments on more extensive, real-world datasets, such as ImageNet, which encompasses a broader range of categories and challenges.
- While diversity is addressed in Table 4, there is a noticeable absence of baseline comparisons with vanilla StyleGAN2 or other diffusion models. Such comparisons would help in understanding whether the method genuinely offers improvements in terms of diversity, or if it simply matches existing capabilities.
- The authors assert that their method solves mode collapse, citing DD-GAN and Diffusion-GAN works. However, these references appear to provide limited empirical evidence, with only DD-GAN offering a toy experiment on the 25-Gaussians dataset. A more comprehensive and substantiated analysis, ideally on a more complex dataset, would bolster the method's credibility in addressing mode collapse.

**Questions:**

- Have the authors considered the potential applicability of the implicit noise schedule produced by their method to other diffusion models? An exploratory comparison with commonly used schedulers in the field could provide valuable insights.
- Is there a discernible difference between implicitly including the noise level in the latent space, as done in the paper, and explicitly passing it like the discriminator?
- Given the application of optimal transport in the paper, do you believe this approach could be extended to regular diffusion models similarly? It could be interesting to explore the generalizability of this idea.

---

> ### Author Response · Authors · 2023-11-21
>
> Thank you for the encouraging remarks about our paper’s contribution which inspire further research, and the valuable feedback from the reviewer. We hope that our responses will solve the reviewer’ questions.
>
> > The NODE map, as evidenced in Figure 6, appears to offer marginal improvements over the vanilla mapping network from StyleGAN2. Consequently, the novelty of the method diminishes, as it seems to reduce to an image and time-conditioned StyleGAN. The authors should address how the method distinguishes itself more significantly from existing approaches.
>
> $\to$ In Fig.6, there is a slight degradation in performance even after replacing the NODE-based mapping network with the StyleGAN2's original mapping network. The StyleGAN2's original mapping network is able to produce good images because the condition $u$ is given to it. In other words, we concatenate the time condition $u$ with input to the StyleGAN2's mapping network.
> However, there is a difference between our NODE-based mapping network and the StyleGAN2’s mapping network. NODE-based mapping networks have homeomorphic characteristics. Because of this homomorphic characteristic, SPI-GAN with a NODE-based mapping network shows an appropriate denoising process when $\hat{\mathbf{i}}(0)$ is generated using $\mathbf{h}(0)$ to $\mathbf{h}(1)$, given a fixed $z$. In contrast, StyleGAN2's mapping network, which has non-homeomorphic properties, does not show a denoising process even when time point $u$ is used as a condition. A visualization of the difference between the NODE-based mapping network and the StyleGAN2 mapping network  is in Figure 10 in Appendix C in our revised paper.
>
> >  The claim of achieving diversity as good as score-based models is made based on datasets that are, in some cases, insufficient for assessing diversity. The resolution of the datasets is often low, and they are primarily single-domain (faces and churches). To validate the diversity claim, it is essential to perform experiments on more extensive, real-world datasets, such as ImageNet, which encompasses a broader range of categories and challenges.
>
> $\to$ Thank you for suggesting an experiment that demonstrates the claim that it achieves as much diversity as a score-based model. However, due to lack of time, we were unable to experiment with ImageNet 64x64, but we will add the results after the discussion period.
>
> >  While diversity is addressed in Table 4, there is a noticeable absence of baseline comparisons with vanilla StyleGAN2 or other diffusion models. Such comparisons would help in understanding whether the method genuinely offers improvements in terms of diversity, or if it simply matches existing capabilities.
>
> $\to$ Thank you for your suggestion. In Table 4, we selected and compared only the best performing models in LSUN-Church-256: Diffusion-GAN, DD-GAN, and CIPs. We will add vanilla StyleGAN2 or other diffusion models as a baseline comparison.
>
> > The authors assert that their method solves mode collapse, citing DD-GAN and The authors assert that their method solves mode collapse, citing DD-GAN and Diffusion-GAN works. However, these references appear to provide limited empirical evidence, with only DD-GAN offering a toy experiment on the 25-Gaussians dataset. A more comprehensive and substantiated analysis, ideally on a more complex dataset, would bolster the method's credibility in addressing mode collapse.
>
> $\to$ We agree that a comprehensive analysis (25-Gaussians datasets) strengthens the reliability of the method in solving the mode collapse problem. However, as reported a couple of times in prior work [1, 2], by showing various (noisy) images (sampled through interpolation), it is expected that the discriminator will not overfit to a subset of clean images.
> [1] Arjovsky, M., & Bottou, L. (2017). Towards principled methods for training generative adversarial networks. The International Conference on Learning Representations
> [2] Xiao, Z., Kreis, K., & Vahdat, A. (2021). Tackling the generative learning trilemma with denoising diffusion gans. arXiv preprint arXiv:2112.07804.

---

> ### Author Response · Authors · 2023-11-21
>
> >  Have the authors considered the potential applicability of the implicit noise schedule produced by their method to other diffusion models? An exploratory comparison with commonly used schedulers in the field could provide valuable insights.
>
> $\to$ Yes, we only focused on imitating diffusion models (especially variance preserve) using GAN, but we expect that the optimal denoising process can be applied to models with different noise diffusion schedules such as NCSN [1] and EDM [2].
>
> [1] Song, Yang, and Stefano Ermon. “Generative modeling by estimating gradients of the data distribution.” Advances in neural information processing systems 32 (2019).
>
> [2] Karras, Tero, et al. “Elucidating the design space of diffusion-based generative models.” Advances in Neural Information Processing Systems 35 (2022): 26565-26577.
>
> > Is there a discernible difference between implicitly including the noise level in the latent space, as done in the paper, and explicitly passing it like the discriminator?
>
> $\to$ The generator appears to include the noise level implicitly. However, since we use the latent vector generated from a NODE-based mapping network, which explicitly includes the noise level, as input to the generator, the generator explicitly includes the noise level. Even if time point $u$ is additionally given to the generator as a condition, the results with SPI-GAN are not significant. We report the performance of this experiment in the following table. This can be confirmed in Table 10 in Appendix E.6 in the revision.
>
> |            CIFAR-10           |  FID |
> |:-----------------------------:|:----:|
> | SPI-GAN (condition $u$ to generator) | 3.03 |
> |            SPI-GAN            | 3.01 |
>
> > Given the application of optimal transport in the paper, do you believe this approach could be extended to regular diffusion models similarly? It could be interesting to explore the generalizability of this idea.
>
> $\to$ Yes, we believe that this approach can be similarly extended to general diffusion models. The diffusion model has been extended to a diffusion model of electromagnetic fields, such as PFGM [1]. We believe that optimal transport can also be applied to various diffusion such as electromagnetic fields.
>
> [1] Xu, Yilun, et al. "Poisson flow generative models." Advances in Neural Information Processing Systems 35 (2022): 16782-16795.

---

> > ### Comment · Reviewer_vo3V · 2023-11-23
> >
> > - **Weakness 1**: Thank you for the visualization on Figure 10 of the NODE-based mapping network denoising, it captures the behavior of both mapping networks nicely. However, even if the generator network can produce correctly noised images, it does not affect the final result as shown in Figure 6, so I still see it as a redundant module.
> > - **Weakness 5**: I apologize for the misunderstanding. My point is that the citations to justify not having mode collapse (DD-GAN and Diffusion-GAN) offer a too simplistic analysis (25-Gaussians). Even including these two citations of Arjovsky et al. and Xiao et al., a more comprehensive analysis is needed to support the claim.
> >
> > Thank you for answering my questions.
> >
> > Unfortunately, my concerns have not been satisfactorily addressed, therefore I maintain my score.

---

> ### Author Response · Authors · 2023-11-22
>
> Dear reviewer vo3V.
>
> We apologize for the late author response during the author/reviewer discussions period. We did our best to conduct additional experiments to help reviewers understand.
>
> Please leave more questions if any. We are ready to answer all your potential unclear points to help your understanding.
>
> Best regards, Authors

---

> ### Author Response · Authors · 2023-11-23
>
> Dear reviewer vo3V
>
> We appreciate the reviewer’s time and effort in reviewing our manuscript and insightful comments.
>
> As the closure of the discussion period is approaching, we would like to bring the review’s attention and check if the reviewer could let us know whether the concerns or the misunderstanding have been addressed by our response.
>
> If this is the case, we would appreciate if you could adjust your rating accordingly.
>
> Best regards, Authors

---

> ### Author Response · Authors · 2023-11-23
>
> We are very grateful for your response.
>
> > Weakness 1: Thank you for the visualization on Figure 10 of the NODE-based mapping network denoising, it captures the behavior of both mapping networks nicely. However, even if the generator network can produce correctly noised images, it does not affect the final result as shown in Figure 6, so I still see it as a redundant module.
>
> $\to$ As you mentioned, there may not be much difference in the final result in Figure 6 between NODE-based mapping and StyleGAN2 mapping networks. However, those are completely different modules, with similar results. Therefore, we expect that, when extended to other advanced diffusion models, the generator network will have a positive impact in correctly generating noisy images.
>
> > Weakness 5: I apologize for the misunderstanding. My point is that the citations to justify not having mode collapse (DD-GAN and Diffusion-GAN) offer a too simplistic analysis (25-Gaussians). Even including these two citations of Arjovsky et al. and Xiao et al., a more comprehensive analysis is needed to support the claim.
>
> $\to$ Due to lack of time we cannot provide further analysis at this time, but it seems that mode collapse has been prevented through the advanced metrics indicating diversity in Table 4. After the discussion period, we will add additional analysis.

---

### Official Review · Reviewer_cKVW · 2023-10-31

**Soundness:** 4 excellent
**Presentation:** 4 excellent
**Contribution:** 3 good
**Rating:** 8
**Confidence:** 3

**Summary:**

SPI-GAN is a denoising diffusion GAN that is a hybrid generative model that is based on GAN with a generator and discriminator and it is trained to denoise the corrupted image. The generator is trained to denoise corrupted images by following a straight path that is the shortest path with minimum Wasserstein distance. And discriminator is a time-dependent discriminator and learns to discriminate images on various interpolation points.

**Strengths:**

The manuscript is well-written and easy to follow.

It explains the gap (trilemma) and addresses the gap with a novel solution. The explanation is supported by formulation and figures.

The evaluations are well-performed and convincing.

They share their code and the trained networks for reproducibility.

**Weaknesses:**

The discussion of the limitations is short, so it can be elaborated.

Although SPI-GAN remodels the task in a simpler way and is easier to learn, its training time is longer.

**Questions:**

Similar to Table 5, can you provide the comparison table for training time? Actually, in the time analyses section, it is said that 'our method only affects the training time' and it is understandable but seeing the difference/effect can be good.

---

> ### Author Response · Authors · 2023-11-21
>
> Thank you for the encouraging remarks about our paper’s contribution and presentation, and the valuable feedback from the reviewer. We hope that our responses will solve the reviewer’ questions.
>
> >  Similar to Table 5, can you provide the comparison table for training time? Actually, in the time analyses section, it is said that 'our method only affects the training time' and it is understandable but seeing the difference/effect can be good.
>
> $\to$ Thank you for the suggestion. The training time for each 1024 CIFAR-10 images is around 32.0s for SPI-GAN and around 45.6s for Diffusion-GAN using four NVIDIA A5000 GPUs. We also revised our paper accordingly.

---

> ### Author Response · Authors · 2023-11-22
>
> Dear reviewer cKVW.
>
> We apologize for the late author response during the author/reviewer discussions period. We did our best to conduct additional experiments to help reviewers understand.
>
> Please leave more questions if any. We are ready to answer all your potential unclear points to help your understanding.
>
> Best regards,
> Authors

---

### Official Review · Reviewer_9sY8 · 2023-11-01

**Soundness:** 2 fair
**Presentation:** 3 good
**Contribution:** 2 fair
**Rating:** 5
**Confidence:** 5

**Summary:**

This paper integrates the diffusion process into Generative Adversarial Networks (GANs) training, striving to achieve the fast image generation capabilities of GANs with the enhanced generation diversity proffered by diffusion models. The authors introduce the diffusion forward process into both the real images and the latent code of the generator, subsequently employing a neural ordinary differential equation (ODE) to facilitate the mapping from $h(0)$ to $h(u)$ for generator's input. With the diffused latent code, the generator is trained to produce images conditioned on time $u$, while the discriminator receives diffused real images. Empirical investigations conducted across three benchmark datasets—CIFAR-10, CelebA-HQ, and LSUN-Church—underscore the approach’s efficacy, showcasing noticeable improvements. Furthermore, the authors demonstrate the generator’s ability in generating images across varied $u$ values.

**Strengths:**

1. Combining diffusion models with GANs for enhanced diversity and a fast generation process presents an intriguing research topic.

2. Employing Neural Ordinary Differential Equations (NODEs) to map embeddings, while conditioning the latent code input to the generator  on time, is an interesting approach.

3. The experimental section demonstrates performance improvements over the baselines.

4. The presentation is clear and easy to follow.

**Weaknesses:**

1. The rationale behind integrating the diffusion process into the generator, particularly in terms of applying it to the latent code input, remains unclear. This might be attributable to the discriminator being exposed to a variety of augmented images, potentially helping to avert overfitting. Meanwhile, the results in Table 3 depict a noticeably inferior performance of SPI-GAN in comparison to both Diffusion-GAN and StyleGANs, raising questions about the soundness of the approach of employing the diffusion process on the generator.

2. An ablation study could be beneficial, exploring alternatives to using the Neural Ordinary Differential Equation (NODE) on the latent code. Investigating the generator’s performance when directly conditioned on the diffused latent code $h(u)$—without the NODE—or simply conditioned on $u$, could provide valuable insights into the viability of such designs.

3. The observed performance gains over the baseline are relatively modest. Particularly in the case of LSUN-Church-256, the proposed method trails behind both Diffusion GANs and StyleGAN2. It's also inconvincing that the paper does not include a comparison of SPI-GAN with DiffusionGAN on CelebA-HQ-256, which could have provided a more comprehensive evaluation.

4. Including details on training time, particularly in comparison to DiffusionGAN, would enhance the paper’s completeness.

**Questions:**

1. Is $u$ applied directly to the image, or is it integrated into an intermediate layer within the discriminator’s architecture?

2. Given that $h(u)$ can be derived from $i(u)$ in the current design, I'm curious to know if the generator has the ability to create images that are very similar to the real images, $i(u)$.

---

> ### Author Response · Authors · 2023-11-21
>
> Thank you for the encouraging remarks about our paper’s contribution and presentation, and the valuable feedback from the reviewer. We hope that our responses will solve the reviewer’ questions.
>
> > The rationale behind integrating the diffusion process into the generator, particularly in terms of applying it to the latent code input, remains unclear. This might be attributable to the discriminator being exposed to a variety of augmented images, potentially helping to avert overfitting. Meanwhile, the results in Table 3 depict a noticeably inferior performance of SPI-GAN in comparison to both Diffusion-GAN and StyleGANs, raising questions about the soundness of the approach of employing the diffusion process on the generator.
>
> $\to$ Our goal is to decrease the large sampling time of the diffusion model by letting a generator learn the straight-path interpolation between latent codes and their corresponding clean images. Given a source and a destination, a Brownian motion used by the diffusion model is inefficient so we propose to **walk** along the straight-path interpolation.
>
> To this end, we first use a neural ODE-based mapping network to convert the straight-path interpolation into the hidden dynamics of $\mathbf{h}(u)$, where $u$ is in $[0,1]$. Therefore, $\mathbf{h}(u)$ contains the information of what to generate at time $u$. Moreover, neural ODEs are theoretically homeomorphic, i.e., continuous and bijective. In Appendix C, we describe the appropriateness of using homeomorphic NODEs for modeling the denoising process of SGMs. For similar purposes, neural ODEs had been already used to learn hidden dynamics of the hidden vectors for videos in [1].
>
> We then train our generator to produce realistic samples from the hidden dynamics of $\mathbf{h}(u)$, where $u$ is in $[0,1]$, being advised by the discriminator. In this way, our generator is able to produce samples along the straight-path interpolation.
>
> [1] Park, Sunghyun, et al. "Vid-ode: Continuous-time video generation with neural ordinary differential equation." Proceedings of the AAAI Conference on Artificial Intelligence. Vol. 35. No. 3. 2021.
>
> >  An ablation study could be beneficial, exploring alternatives to using the Neural Ordinary Differential Equation (NODE) on the latent code. Investigating the generator’s performance when directly conditioned on the diffused latent code ℎ(u)—without the NODE—or simply conditioned on u, could provide valuable insights into the viability of such designs.
>
> $\to$ In Figure 6, we have the result of deriving h(u) by using the latent code and $u$ as input to the StyleGAN2 mapping network without NODE. Additionally, as you suggest, we report the results of simply applying $u$ as a condition to the generator, without a mapping network in the following table. The following table shows that a mapping network is essential for SPI-GAN. This can be confirmed in Table 9 in Appendix E.6 in the revision.
>
> |            CIFAR-10           |  FID |
> |:-----------------------------:|:----:|
> | SPI-GAN (w/o mapping network) | 5.72 |
> |            SPI-GAN            | 3.01 |
>
> > The observed performance gains over the baseline are relatively modest. Particularly in the case of LSUN-Church-256, the proposed method trails behind both Diffusion GANs and StyleGAN2. It's also inconvincing that the paper does not include a comparison of SPI-GAN with DiffusionGAN on CelebA-HQ-256, which could have provided a more comprehensive evaluation.
>
> $\to$ In LSUN-Church-256, SPI-GAN is inferior to Diffusion-GAN and StyleGAN in terms of FID performance, but please note that it shows superior to Diffusion-GAN and CIPs (State-of-the-art model in LSUN-Church-256) in terms of Recall and Coverage, which are recently proposed advanced metrics.
>
> > Including details on training time, particularly in comparison to DiffusionGAN, would enhance the paper’s completeness.
>
> $\to$ Thank you for the suggestion. The training time for each 1024 CIFAR-10 images is around 32s for SPI-GAN and around 45.6s for Diffusion-GAN using four NVIDIA A5000 GPUs. We also revised our paper accordingly.

---

> ### Author Response · Authors · 2023-11-21
>
> >  Is u applied directly to the image, or is it integrated into an intermediate layer within the discriminator’s architecture?
>
> $\to$ $u$ is the temporal condition of the generated image and is used as input to the discriminator. In other words, $u$ is concatenated with the generated image and used as input to the discriminator. Using this concatenated information, the discriminator is able to know the temporal condition of input images.
>
> > Given that $ℎ(u)$ can be derived from $i(u)$ in the current design, I'm curious to know if the generator has the ability to create images that are very similar to the real images, $i(u)$
>
> $\to$ As your comment, we compare $\hat{\mathbf{i}}(u)$ derived from $\mathbf{h}(u)$ and the ground truth $\mathbf{i}(u)$ in Figures 15 in the Appendix of our revised paper. As a result, it appears that $\mathbf{i}(u)$ and $\hat{\mathbf{i}}(u)$ have similar distributions.

---

> ### Author Response · Authors · 2023-11-22
>
> Dear reviewer 9sY8.
>
> We apologize for the late author response during the author/reviewer discussions period. We did our best to conduct additional experiments to help reviewers understand.
>
> Please leave more questions if any. We are ready to answer all your potential unclear points to help your understanding.
>
> Best regards,
> Authors

---

> ### Author Response · Authors · 2023-11-23
>
> Dear reviewer 9sY8
>
> We appreciate the reviewer’s time and effort in reviewing our manuscript and insightful comments.
>
> As the closure of the discussion period is approaching, we would like to bring the review’s attention and check if the reviewer could let us know whether the concerns or the misunderstanding have been addressed by our response.
>
> If this is the case, we would appreciate if you could adjust your rating accordingly.
>
> Best regards, Authors

---

### Author Response · Authors · 2023-11-21

Dear All Reviewers,

We thank the reviewers for taking the time to read, evaluate, and provide valuable feedback. We revised the following points and uploaded a new version:

1. We described the effectiveness of the neural ordinary differential equations-based mapping network in Appendix C.
2. We added a training time in Appendix E.5.
3. We also added additional experimental results of ablation studies in Tables 9 & 10 in Appendix E.6.
4. We added visualization of generated image $\hat{\mathbf{i}}(u)$ and noise image $\mathbf{i}(u)$ in Appendix F.4.
5. We revised Figure 7 and its caption in revision.

Best regards,

Authors

---

### Meta-Review · Area_Chair_Xv6W · 2023-12-06

**Metareview:**

3x BR and 1x A. This paper proposes to bridge between GAN and diffusion models in order to achieve both efficient and covering generation. The reviewers agree on the (1) clear presentation, (2) intriguing topic, and (3) interesting approach. However, they also have common concerns about the (1) unclear motivation, (2) insufficient testing datasets (low resolution, limited domains), and (3) marginal improvement or worse performance. The rebuttal does not address their concerns, and the only “accept” reviewer’s comments are short and not active. The AC therefore leans not to accept this submission.

**Justification For Why Not Higher Score:**

N/A

**Justification For Why Not Lower Score:**

N/A

---

### Decision · Program_Chairs · 2024-01-16

Reject